# Heterogeneous run-and-tumble motion accounts for transient non-Gaussian super-diffusion in haematopoietic multi-potent progenitor cells

Benjamin Partridge[1], Sara Gonzalez Anton[2,3], Reema Khorshed[2], George Adams[2,3], Constandina Pospori[2,3], Cristina Lo Celso[2,3] *, Chiu Fan Lee[1] *

1 Department of Bioengineering, Imperial College London, South Kensington Campus, London, United Kingdom, 2 Department of Life Sciences, Imperial College London, South Kensington Campus, London, United Kingdom, 3 Sir Francis Crick Institute, London, United Kingdom

* c.lo-celso@imperial.ac.uk (CLC); c.lee@imperial.ac.uk (CFL)

## Abstract

Multi-potent progenitor (MPP) cells act as a key intermediary step between haematopoietic stem cells and the entirety of the mature blood cell system. Their eventual fate determination is thought to be achieved through migration in and out of spatially distinct niches. Here we first analyze statistically MPP cell trajectory data obtained from a series of long time-course 3D *in vivo* imaging experiments on irradiated mouse calvaria, and report that MPPs display transient super-diffusion with apparent non-Gaussian displacement distributions. Second, we explain these experimental findings using a run-and-tumble model of cell motion which incorporates the observed dynamical heterogeneity of the MPPs. Third, we use our model to extrapolate the dynamics to time-periods currently inaccessible experimentally, which enables us to quantitatively estimate the time and length scales at which super-diffusion transitions to Fickian diffusion. Our work sheds light on the potential importance of motility in early haematopoietic progenitor function.

## I. Introduction

The haematopoietic system is responsible for the generation of billions of new blood cells daily. This considerable feat is orchestrated from the bone marrow whose constituent blood cells are organized into a hierarchical lineage tree, atop of which reside haematopoietic stem cells (HSCs). HSCs have two defining properties; self-renewal—the ability to replenish their own numbers, and multi-potency—the potential to differentiate into any given blood cell type. Downstream of HSCs lie multi-potent progenitor cells (MPPs), which act as an intermediary between HSCs and mature blood cells. Their successive proliferation and differentiation amplifies cell numbers to enable a small pool of HSCs to achieve a staggering output of mature blood cells [1].

**Data Availability Statement:** Relevant data and code are held in the following public repository: DOI:10.5281/zenodo.7019736.

**Funding:** B.P., BB/M011178/1, Biotechnology and Biological Sciences Research Council, https://bbsrc.ukri.org/ S.G.A., C36195/A27830, Cancer Research UK, https://www.cancerresearchuk.org/ C.P., 15040, Blood Cancer UK, https://bloodcancer.org.uk/ C.L.C., 15040, Blood Cancer UK, https://bloodcancer.org.uk/ C.L.C., BB/L023776/1, Biotechnology and Biological Sciences Research Council, https://bbsrc.ukri.org/ C.L.C., C36195/A26770, Cancer Research UK, https://www.cancerresearchuk.org/ C.L.C., 212304/Z/18/Z, Wellcome Trust, https://wellcome.org/ None of the funders mentioned above play any role in the study design, data collection and analysis, decision to publish, or preparation of the manuscript.

**Competing interests:** The authors have declared that no competing interests exist.

To achieve tissue homeostasis and to properly respond to stress and infection cell division and differentiation must be tightly regulated. The motility of haematopoietic stem and progenitor cells (HSPCs) is emerging as an unexpected component to these regulations [2]. Therefore, a detailed understanding of the spatial organization and dynamics underpinning the relationship between HSPCs and their environment is crucial. Intravital microscopy has been a key experimental technique that directly probes this relationship [3, 4]. This is of special clinical interest in the irradiated state, due to the importance of radiation therapy in conditioning the bone marrow prior to transplantation when treating malignancies of the blood.

In parallel development, the diffusive properties of migrating organisms have been the subject of intense interest within the statistical physics community [5, 6]. A common characteristic amongst such systems is a non-linear, power-law scaling of the mean square displacement (MSD) with time. This phenomenon stands in contrast to the linear relationship expected in the 'typical' case of Fickian diffusion. This 'anomalous' diffusion—described as super-diffusion (sub-diffusion) in cases where the MSD growth exceeds (falls below) the linear case—has been postulated to influence a variety of biological processes spanning virtually all biological length-scales [7–21]. Two recent intravital imaging studies have examined the dynamical behavior of HSPCs in their native, steady-state environment and found that i) progenitor cells display enhanced motility relative to HSCs, and ii) temporal heterogeneity within HSC trajectories, where the dynamical behavior alternated between periods of a confined random walk and processive motion [2, 4]. There are a limited number of prior works investigating the migratory behavior of transplanted HSCs [22, 23], and to our knowledge a quantitative analysis of the diffusive properties of haematopoietic MPPs in an irradiated setting has yet to have been undertaken.

In this article, we first report on a statistical analysis of cell trajectory data taken from 3D *in vivo* imaging experiments of haematopoietic multi-potent progenitor cells in the irradiated bone marrow cavity of murine calvaria (Fig 1). Many of the cell trajectories are observed over long time periods atypical of 3D *in-vivo* bone marrow imaging experiments, with 44% of trajectories having a length greater than 3 hours and 17% having a length of greater than 6 hours. We demonstrate that the cells display transient non-Gaussian super-diffusion over time-scales of biological interest. We then explain this observation using a data-driven run-and-tumble

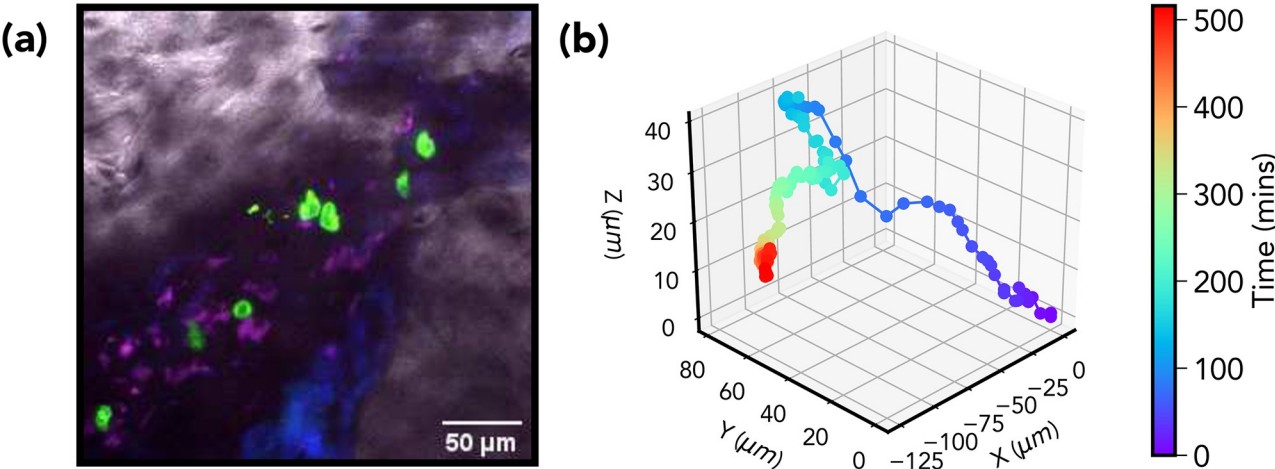

**Fig 1.** (a) Haematopoietic multi-potent progenitor cells (green) and the local micro-environment in an irradiated bone marrow cavity taken from a time-lapse in vivo imaging experiment (grey: bone; blue and purple: autofluorescence). (b) 3D reconstruction of a MPP trajectory extracted from the same set of experiments.

(RTM) model which takes into account heterogeneity in the dynamics of the ensemble. We found that the incorporation of heterogeneity into our RTM is necessary and sufficient to explain the non-Gaussian super-diffusive behavior. We extrapolate the dynamics to time-periods currently inaccessible to time-lapse imaging experiments which enables us to quantitatively estimate the time and length scales at which super-diffusion transitions to Fickian diffusion. These estimates will be integral to understanding how stem and MPP cell motility influence the regulations of blood cell generation due to the recognized importance of spatial organization in controlling the function of the haematopoietic stem cell niche.

## II. Results

### A. Multi-potent progenitor cells display non-gaussian super-diffusion

The cell trajectory data presented were extracted from a series of *in-vivo* imaging experiments in which labeled multi-potent progenitor cells expressing membrane-bound green fluorescence protein (GFP) purified from the bone marrow of donor mice are transplanted into myeloablative recipient mice and their three-dimensional positions followed using confocal microscopy two days after radiation therapy for time periods ranging from 18 to 525 minutes with three-minute intervals between subsequent frames (see Materials and methods). In Fig 1(a) we show a maximum *z*-projection still frame depicting a typical configuration of such an experiment with a few haematopoietic MPP cells (green) and the local micro-environment. A representation of an extracted MPP trajectory is shown in Fig 1(b).

The standard indicator of anomalous diffusion is a power-law scaling of the mean square displacement

$$\langle |\mathbf{r}(t)|^2 \rangle \propto t^\alpha \tag{1}$$

where the angular brackets indicate an ensemble average. An exponent of $\alpha > 1$ indicates super-diffusion, which is the regime of interest here. The underlying mechanism responsible for this result depends upon the nature of the stochastic process generating the motion. For instance, a continuous-time random walk in which the displacement of random walker moving at a fixed speed is given by

$$\mathbf{r}(t) = \sum_{t=0}^{T} \Delta \mathbf{r}(t) \tag{2}$$

where $\Delta\mathbf{r}$ are run lengths represented by random variables distributed according to $P(\Delta\mathbf{r}, t)$. In the limit of large $T$, assuming independent run lengths with finite variance, the Central Limit Theorem (CLT) dictates that the distribution Prob($\mathbf{r}$, $t$) of the ensemble (also known as the propagator) will converge to a Gaussian distribution with a second moment that scales linearly with time. Super-diffusion occurs when the assumptions of the CLT are violated. Models with heavy-tailed step length distributions, such as Levy walks [24], violate the finite variance assumption. Whilst models with persistent correlation, such as the Elephant walk [25, 26], violate the assumption of statistically independent displacements. Furthermore, a number of studies [9, 11, 12] have shown empirically that a useful investigative tool is the scaling property

$$\text{Prob}(\mathbf{r}, t) = t^{-\beta} F\left(\frac{\mathbf{r}}{t^\beta}\right) \tag{3}$$

where $F$ is generally non-Gaussian and $\beta > 0.5$.

Due to the limited amount and length of trajectory data typically collected during *in vivo* imaging experiments direct access to the ensemble averaged MSD is not possible. Thus in order to improve the statistical properties of the estimator the time-averaged mean square displacement is computed instead (TAMSD) [5].

$$\overline{\delta_i^2(\Delta)} = \frac{1}{T - \Delta} \int_0^{T-\Delta} |\mathbf{r}(t + \Delta) - \mathbf{r}(t)|^2 \mathrm{d}t \tag{4}$$

The subscript $i$ indicates the individual cell in question and the over-script bar the time average. This quantity may then be further averaged over all $N$ tracks

$$\left\langle \overline{\delta^2(\Delta)} \right\rangle = \frac{1}{N} \sum_{i=0}^{N} \overline{\delta_i^2(\Delta)}. \tag{5}$$

In Fig 2 we present the evidence for transient MPP super-diffusion. Fig 2(a) shows the the power-law scaling (plotted in log-log scale) of the TAMSD averaged over all cells. In this figure we have taken a maximum lag-time of two and a half hours, and to ensure a reasonable statistical convergence for later time points we have used tracks of three hours or more in length. An ordinary least-squares fit yields a gradient of $\alpha = 1.22$, indicative of a super-diffusive scaling.

In Fig 2(b) we plot the probability distributions of the re-scaled variable $\eta = \delta x/\Delta^\beta$ for various lag times $\Delta$, where $\delta x = x(t + \Delta) - x(t)$; $x(t)$ being the $x$-coordinate of a given cell at time $t$. The value of the exponent $\beta$ was determined to be 0.72 using a procedure described in Materials and Methods (Fig 6). A similar result was observed for the $y$-coordinate, however, the $z$-coordinate showed a lesser value of 0.59—this is likely an artifact of the limited field of view in the $z$-direction, a common limitation of 3D intravital imaging experiments. The resulting curve-collapse demonstrates the super-diffusive scaling property expressed in Eq (3) over a three hour time window. It is also of note that shape of the distributions appears to be

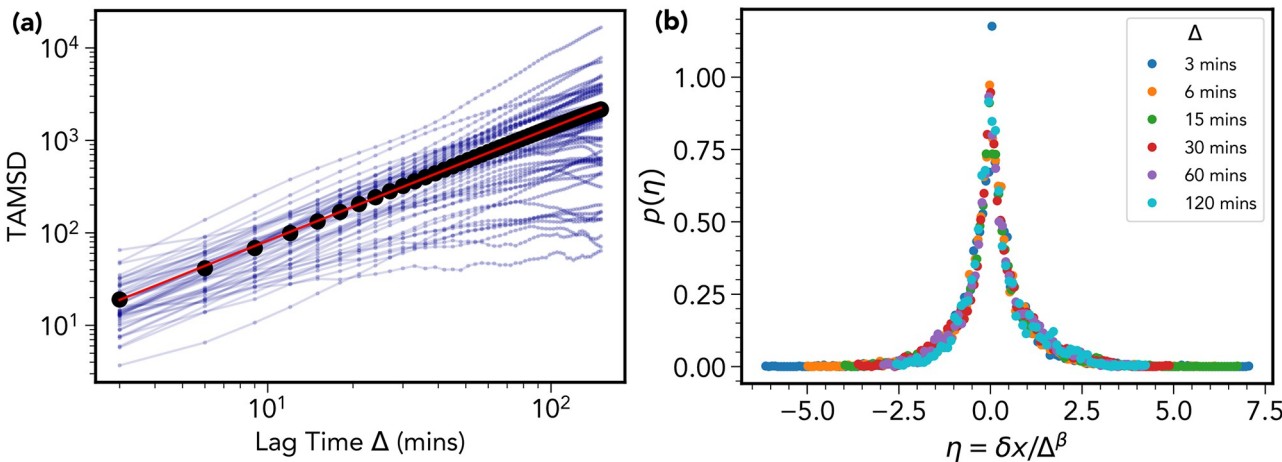

**Fig 2. Multi-potent progenitor cells display transient non-Gaussian super-diffusion.** (a) Time-averaged mean square displacement trajectories for each cell of track length greater than 3 hours (blue curves) along with the average over all cells (black curve). A linear fit (red curve) yields a super-diffusive exponent of $\alpha = 1.22$. (b) Curve collapse of the distribution of the re-scaled variable $\eta = \delta x/\Delta^\beta$, where $\delta x = x(t + \Delta) - x(t)$, for 6 different lag time intervals $\Delta$. The value of the exponent $\beta$ is estimated to be 0.72 using a procedure described in Sec. IV.. For standard Fickian diffusion it is expected that $\alpha = 1$ and $\beta = 0.5$.

distinctly non-Gaussian, a characteristic also quantified in Materials and Methods (see Figs 7 and 8).

## B. Run and tumble model explains transient super-diffusion

To explain the apparent transient super-diffusion displayed by the MPP population we implement a run and tumble (RTB) model of cell motility. RTB models have been successful in the dynamical description of a variety of motile organisms including bacteria *E. coli* and Salmonella [27, 28], and the eukaryotic unicellular alga Chlamydomonas [29, 30]. A random walker undergoes intermittent linear 'runs' of length $r$ for a time-interval $\Delta t$ before re-orientating its direction by angle $\theta$ and proceeding to embark on another run. In the simplest case $r$ and $\Delta t$ are related through a fixed speed $V$. This case would not provide a faithful representation of our data-set due to internal heterogeneity within the cell trajectories. Our model requires the specification of the run length distribution $f(r)$, the turn angle distribution $h(\theta)$, and the run time distribution $g(\Delta t)$.

To avoid the arbitrary imposition of a parametric model onto the data-set we take a data driven approach, in which the relevant distributions specifying the model correspond to empirical distributions constructed from the MPP data-set. This allows for the avoidance of model error in our inference. We discuss the details of our approach in Materials and Methods, while we briefly describe our procedure here. The empirical distributions are constructed by coarse-graining each trajectory to include points with a threshold value of at least 7.2 microns (approximately one cell diameter) separation. The threshold value was determined as described in Materials and Methods. This procedure is done to remove short-range, transient fluctuations in the cell centroid position unrelated to translocations of the entire cell body. Small changes to the threshold value do not significantly alter the results. Displacements between points on these coarse-grained trajectories enable the definition of run length, run time, and turn angle distributions. In Materials and Methods, we further implement the more elaborate Bayesian method of [31] to complement and to confirm the results of the minimal RTB model presented herein.

We simulate a trajectory using the following procedure which is described in detail in Materials and Methods:

1. Assign average run length $\overline{r}$, run time $\overline{t}$ and average turn angle $\overline{\theta}$.

2. Draw from scaled run length distribution $f(r/\overline{r})$ and multiply by $\overline{r}$ to obtain current run-length.

3. Draw from scaled turn angle distribution $g(\theta/\overline{\theta})$ and multiply by $\overline{\theta}$ to obtain current turn angle.

4. Draw from scaled run time distribution $h(\Delta t/\overline{\Delta t})$ and multiply by $\overline{t}$ to obtain current run-time.

5. Update time and position variables and return to 2.

Note that our model incorporates the observed inter-track heterogeneity. This is manifested via significant inter-trajectory variation in the mean run length and run-time and mean turn angle. We account for this by drawing from scaled empirical distributions. For example, each experimental run-time interval is scaled by the average for the corresponding cell trajectory. These dimensionless values are then aggregated across all trajectories to obtain the model empirical distribution function. In other words, for each trajectory we have a set of run times $\{\Delta t_1, \Delta t_2, \ldots, \Delta t_T\}$ and an associated mean run-time $\overline{\Delta t}$. For each simulated track we assign a

$\overline{\Delta t}$ from the list of empirical mean run-times and then generate (dimensionless) samples from the distribution $h(\Delta t / \overline{\Delta t})$. We then obtain a true run-time in minutes through multiplication by the corresponding scale factor $\overline{\Delta t}$. Run length and angular heterogeneity was incorporated by applying the same procedure. Heterogeneous diffusion processes have been implicated in the emergence of non-Gaussian displacement distributions [31–35]. Fig 3 shows that our data-driven model accounts for the super-diffusive behavior observed in the data. In Materials and Methods we demonstrate that the ensemble of simulated trajectories display non-Gaussian displacement distributions similar to those computed from the data.

Beyond explaining the experimental findings, our simulation model also enables us to investigate MPP dynamics at timescales inaccessible to intravital microscopy experiments. The curve is the result of simulating 51 trajectories, one for each average run-time calculated from the trajectory data, for 30000 minutes, and the average taken using a maximum lag-time of 1500 minutes, an order of magnitude greater than the empirical data. We can infer by inspection of this result that the transition to Fickian diffusion takes place at around the 100-200 minute mark. If we refer to the most exploratory cell in Fig 2(a) we can then infer an approximate upper limit to the spatial region covered by the persistent motion to be at least $100\mu m$ or around 20 cell diameters in length.

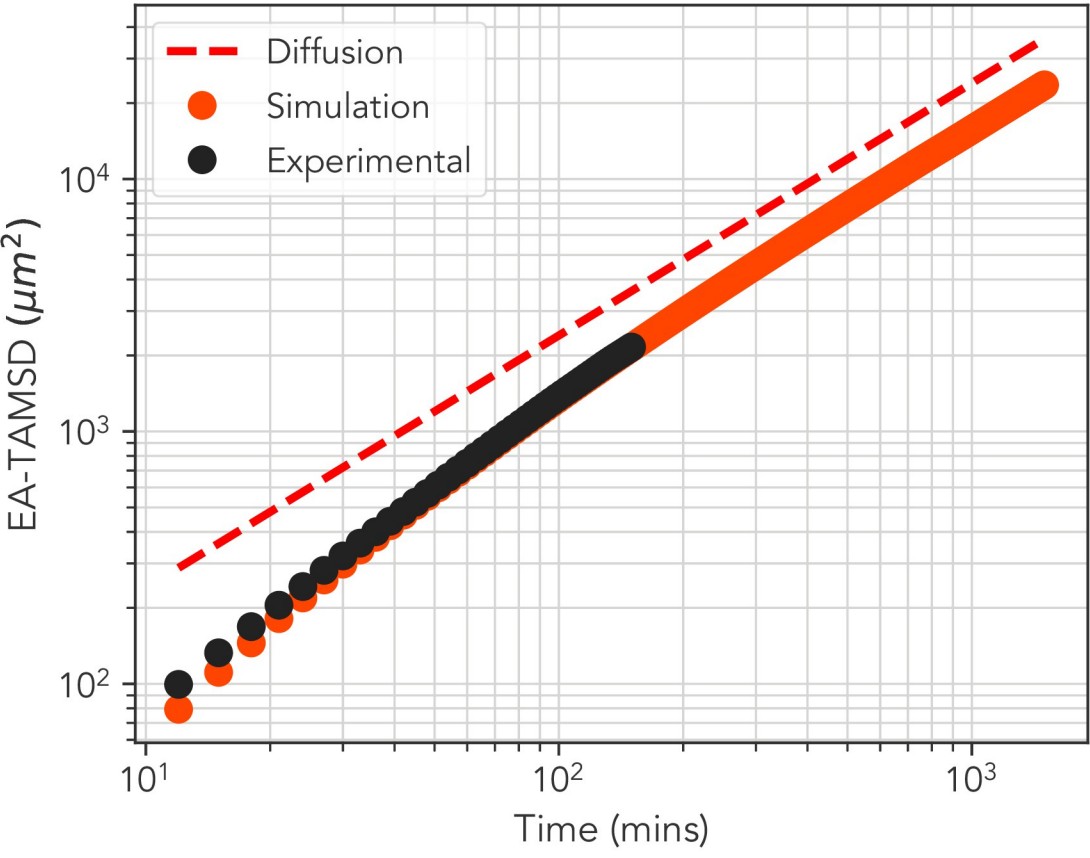

**Fig 3. Comparison of simulation results and experimental data.** Ensemble averaged TAMSD curves for experimental (black) and simulation (orange) alongside the line of diffusion (red dashed), which is included as a guide to the eye. The simulation data has been extended into a region inaccessible to experimentation in which a crossover to diffusion is observable. Error bars are too small to be visible.

## III. Discussion

### A. Heterogeneity

We have demonstrated that the haematopoietic multi-potent progenitor cells display transient super-diffusion which can be explained by a run-and-tumble model incorporating one necessary and sufficient feature: heterogeneity between cell tracks. The heterogeneity present within the cell motion is notable, and in line with the phenotypic and functional heterogeneity observed in the biological literature [36]—there is significant variation in the average run-time and turn-angle for each cell, evidently giving rise to a spectrum of diffusivities.

Our results did not show any evidence to suggest that the transient super-diffusion observed in the MPP trajectory data could be attributed to heavy-tailed run length distributions, precluding a Levy Walk style model. Instead, incorporation of heterogeneity in run-length, turn-angle, and run-time sufficiently re-capitulated the experimental TAMSD.

Indeed, heterogeneity has been implicated as an important factor contributing to the anomalous statistical behavior in populations of motile cells [5, 31, 37]. A number of theoretical and empirical studies have emerged demonstrating that the canonical indicators of super-diffusion; namely a scaling of the mean square displacement $\sim t^\alpha$ with $\alpha > 1$ and non-Gaussian displacement distributions can originate from simple models of heterogeneous persistent cell motion [31, 35, 37, 38].

In particular, mathematical models incorporating heterogeneity have found particular use in explaining the so-called "Brownian yet non-Gaussian diffusion" in systems displaying a normal linear scaling of the mean square displacement with time, while having non-Gaussian step width distributions [33, 34, 39–41]. It is noted [34] that this is facilitated by the lack of a strict separation of time-scales between the slowly varying heterogeneity and the onset of Fickian diffusion.

Pertinently, heterogeneity has also been postulated as a putative mechanism for super-diffusivity over long times in experimental tracking of ensembles of motile cells [31, 32, 35, 37, 38, 42, 43]. This stands as an alternative to the Lévy walk model, which postulates a power law step length distribution, with uniformly distributed turn angles between steps [8, 44–46]. This point is highlighted in a recent paper [38] where the authors compare a heterogeneous run-and-tumble model to a Lévy walk model as a mechanism for explaining the experimentally observed super-diffusive scaling of mouse fibroblast cells. The authors conclude that heterogeneous run and tumble motion provided a superior explanation for the observed super-diffusive scaling.

Heterogeneity in the dynamical behavior of motile cells has numerous causes. Some of these are internal to the cell, reflecting variation in the transcriptional states of the underlying genome which pertain to cellular motility. Others are induced by environmental differences in the cells immediate surroundings, giving rise to a spatially dependent diffusion coefficient $D(\mathbf{r})$. The latter scenario is especially relevant in *in-vivo* settings such as the bone marrow cavity, which as a physical medium is known for its complex heterogeneity. In *in-vitro* settings, environmental heterogeneity is directly controlled for, hence it is more natural to assume that each cell trajectory is generated by an identical probabilistic model. Following Ref. [35], it is therefore useful to outline two limiting cases which categorize heterogeneity within a cell population.

**1. Temporal heterogeneity.** Temporal heterogeneity refers to statistical inhomogeneity with a single observed cell trajectory, and is typically manifested as stochastic alternation between different dynamical modes. For example, periods of inactivity where the cell centroid is effectively stationary are followed by bursts of motility. A recent paper on 2D super-diffusive behavior observed in a population on motile ameoba [11] noted this characteristic in their

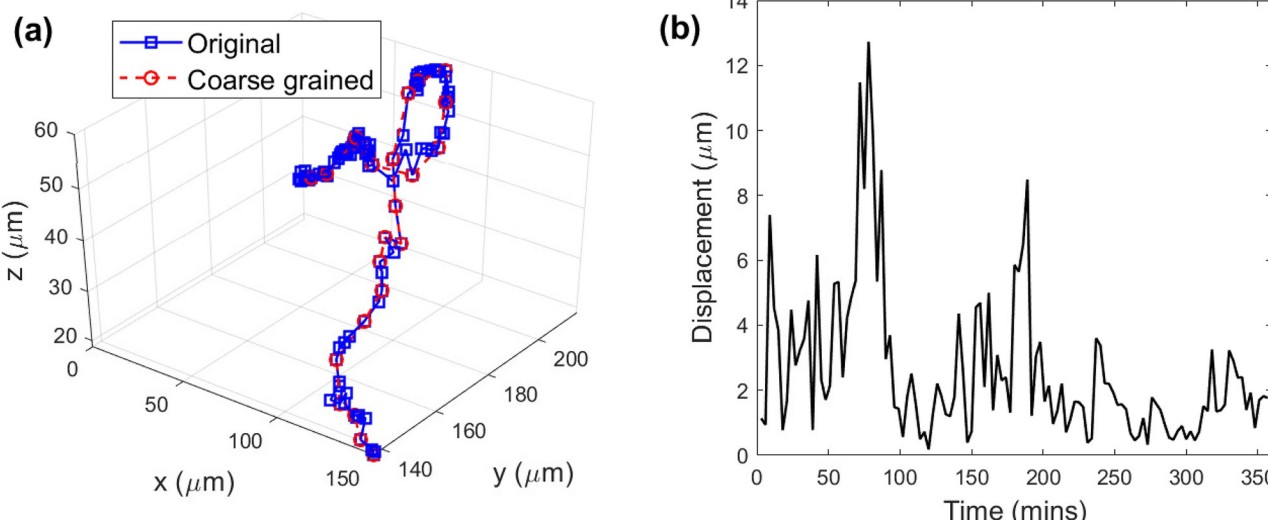

**Fig 4. Temporal heterogeneity.** (a) shows an example coarse-grained track (red) superimposed onto it's original (blue) in which each subsequent pair of points are separated by 3 minute intervals. Visually it appears as though periods of low motility and persistence are followed by more exploratory periods displaying persistent motion. During periods of lessened motility sampled displacements may not faithfully represented true translocation of entire cell; the coarse-graining procedure described in Sec. IV lessens the effect of this heterogeneity. The displacement length time-series of the original track is shown for the same trajectory in (b).

motion. During periods of relative inactivity the sampled positions of the cell centroid will not represent true translocations of the entire cell, but fluctuations of the centroid position caused by transient cell surface membrane fluctuations, which themselves are likely to be driven by re-arrangements of the underlying cellular cytoskeleton. Fig 4(a) shows an example of an MPP trajectory along with its corresponding displacement time-series in (b). Qualitatively one may discern periods of very slow diffusion, followed by stretches of persistent motion.

**2. Cellular heterogeneity.** Cellular heterogeneity reflects time-independent, inter-cell variation in the statistical behavior of the cell trajectories. The origin of such variation is likely due to variation in the environment within a given bone marrow cavity. This can be explained through variation in the mean run time, and variation in the mean turn angle (as shown in the insets of Fig 12(b)–12(d) in Materials and methods). Such inter-cell variation accounts for the significant spread in TAMSD curves shown in Fig 2(a).

## B. Transition to Fickian diffusion

Our model-facilitated extrapolation point to an upper limit on the range of influence of the directed motion of MPPs within the bone marrow cavity of irradiated murine calvaria. We infer this limit from the cross-over to Fickian diffusion of the simulated TAMSD. The high value of the upper limit is surprising in part because it spans several multiples of cell diameter, suggesting that persistent motion may play a significant role in determining the biophysical properties of early stage bone marrow tissue regeneration. A key concept within haematopoietic stem cell biology is that of a niche—a distinct anatomical compartment within the bone marrow whose cellular constituents directly regulate cell fate. Whilst our study makes no direct connection between potential niches in the bone marrow and MPP motility, it may be reasonable to suggest that the persistent motility observed could serve as a stochastic mechanism through which progenitor cells are able to re-locate to regions within close proximity to potential niches.

In summary, we have employed a data-driven RTM to explain the non-Gaussian, super-diffusive dynamics of MPP observed in long time-course 3D *in vivo* imaging experiments on irradiated murine bone marrow, and used our model to quantify the temporal and spatial ranges in which these anomalous features are present. Interesting future directions will be to connect these anomalous dynamics to bone marrow regeneration, niche organization, and homeostasis.

## IV. Materials and methods

### A. Experimental description

Bone marrow 3D intravital imaging experiments aim to selectively track the positions of cells in a given subset of the haematopoietic lineage tree in order to observe their dynamical behavior. Such information is crucial to a complete understanding of the processes which underpin haematopoietic stem and progenitor cell (HSPC) biology; allowing insight into cell motility, cell division and apoptosis, and their relation to the bone marrow micro-environment.

Haematopoietic stem cells (HSCs) and multi-potent progenitor cells (MPPs) are defined operationally through their ability to provide long- and sort-term reconstitution capacities, respectively, to the haematopoietic tissue. However, identifying HSCs using such a definition requires the use of serial transplantation assays, clearly limiting the ability to perform experiments to visualize HSCs and related populations *in-vivo*. Instead, we identify haematopoietic cell populations phenotypically using cell surface marker proteins. For example, haematopoietic progenitors may be identified by selecting populations of cells lacking the expression of a cocktail of cell surface markers associated with terminal differentiation—termed lineage negative (Lin-). This population may then be enriched further for markers associated with self-renewal capacity and multi-potency.

To visualize haematopoietic progenitor cells *in-vivo* using confocal or multi-photon microscopy, cells must be fluorescently labeled. There exist a number of strategies through which this is possible. The experiments from which the data used in this letter was extracted transplant and fluorescently labeled multi-potent progenitor (MPP) cells into an irradiated recipient. Fluorescent haematopoietic cells are extracted from mice genetically modified to express a florescent protein in their haematopoietic cells. Such cells are then purified into a rare population of multi-potent progenitor cells using fluorescence activated cell sorting to identify their associated cell surface markers. Fig 5 shows a schematic of this process.

**1. Mice.** All animal work was conducted as regulated by the UK Home Office, under project licence PP9504146 approved by the Imperial College's Animal Welfare and Ethical Review Body (AWERB) committee and the UK Home Office. Recipient mice are irradiated using two doses of 5.5Gy of $\gamma$-radiation three hours apart. This procedure is to condition the mice to

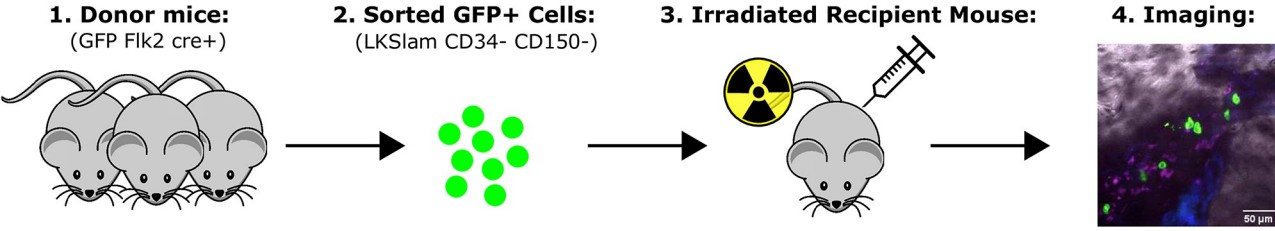

**Fig 5. Experimental setup.** Schematic showing the experimental setup used to generate the data used in this article. Bone marrow harvested from transgenic donor mice expressing a fluorophore protein in their haematopoietic cells is purified using fluorescence activated cell sorting (FACS) to produce a sub-set of bone marrow cells highly enriched for multi-potent progenitors (MPPs). These cells are then transplanted into syngeneic recipient mice and imaged using confocal microscopy two days after the transplantation.

make them receptive of HSPC engraftment. The mice are then culled at the end of the experiment (within 2-4 days from irradiation and HSPC administration) using cervical dislocation as required by the Home Office Schedule 1 of the Animals (Scientific Procedures) Act 1986, amended 2012.

**2. Cell sorting.** The following set of cell surface markers were used to identify MPP cells: Lineage-, c-Kit+, Sca-1+, CD150-, CD48+. A '+' sign preceding the marker name indicates that the cell population is enriched for that particular marker, likewise a '-' sign indicates depletion.

- Lin-: a cocktail of markers indicating terminally differentiated haematopoietic cells, namely CD3, CD4, CD8, B220, CD11b, Gr-1, and Ter119.

- Stem cell antigen Sca1+.

- Stem cell growth factor cKit+.

- CD48+ and CD150-.

**3. Microscopy.** Laser scanning confocal microscopy (LSCM) was used to record time-lapse images of the motion of MPP cells observed at various fixed positions within the calvarium bone marrow. Imaging commenced two days after transplantation was performed.

**4. Data acquisition.** After the time-lapse images were recorded the central positions of the cell were extracted in a semi-automated fashion using IMARIS software.

## B. Details of statistical analysis

**1. Sensitivity analysis of curve collapse.** The value of the exponent $\beta$ in Eq (3) in Sec. IIA was determined to be 0.72 using the following procedure. For each value of the lag times $\Delta = \{3, 6, 15, 30, 60, 120\}$(mins), we compute the Wasserstein distance $l(u, v)$: The Wasserstein distance for two random variables with CDFs $U$ and $V$ is given by

$$l(u, v) = \int_{-\infty}^{\infty} |U - V| \, \mathrm{d}u\mathrm{d}v \ . \tag{6}$$

The value of the exponent $\beta$ is chosen as the one which minimizes the maximum value of $l(u, v)$ among the distributions. The result is displayed in Fig 6, from which a clear minimum at $\beta = 0.72$ can be observed, which is the value used to produce the curve collapse in Fig 2(b).

**2. Quantification of non-gaussianity.** Following on from the method of [11, 12] we attempt to quantify the non-Gaussian nature of the MPP displacement distributions over several lag times $\Delta$. To do this, for lag-times $\Delta = \{3, 6, 15, 30, 60, 120\}$ mins we plot distributions of the lagged $x$-displacement $\delta x = x(t + \Delta) - x(t)$ for all values $t \in [0, T - \Delta]$ where $T$ is the temporal length of a given track. We include all tracks long enough to produce at least one value of $\delta x$ for the largest lag time $\Delta_{max} = 120$ mins. Using maximum likelihood estimation (MLE) (implemented using SciPy [47]), we then estimated the parameters of a generalized Gaussian distribution

$$f(\delta x; \gamma, \sigma) = \frac{\gamma}{2\sigma^2 \Gamma(1/\gamma)} \exp(|\delta x/\sigma|^{\gamma}) \tag{7}$$

where $\sigma$ is the scale parameter of the distribution, and $\Gamma(x)$ the Gamma function, required for proper normalization of the distribution. The shape parameter $\gamma$ dictates the tailedness of the distribution—the greater it's value the more likely there is to be larger displacements. For $\gamma = 2$ the standard normal distribution is recovered, for $\gamma = 1$ a Laplacian distribution is realized.

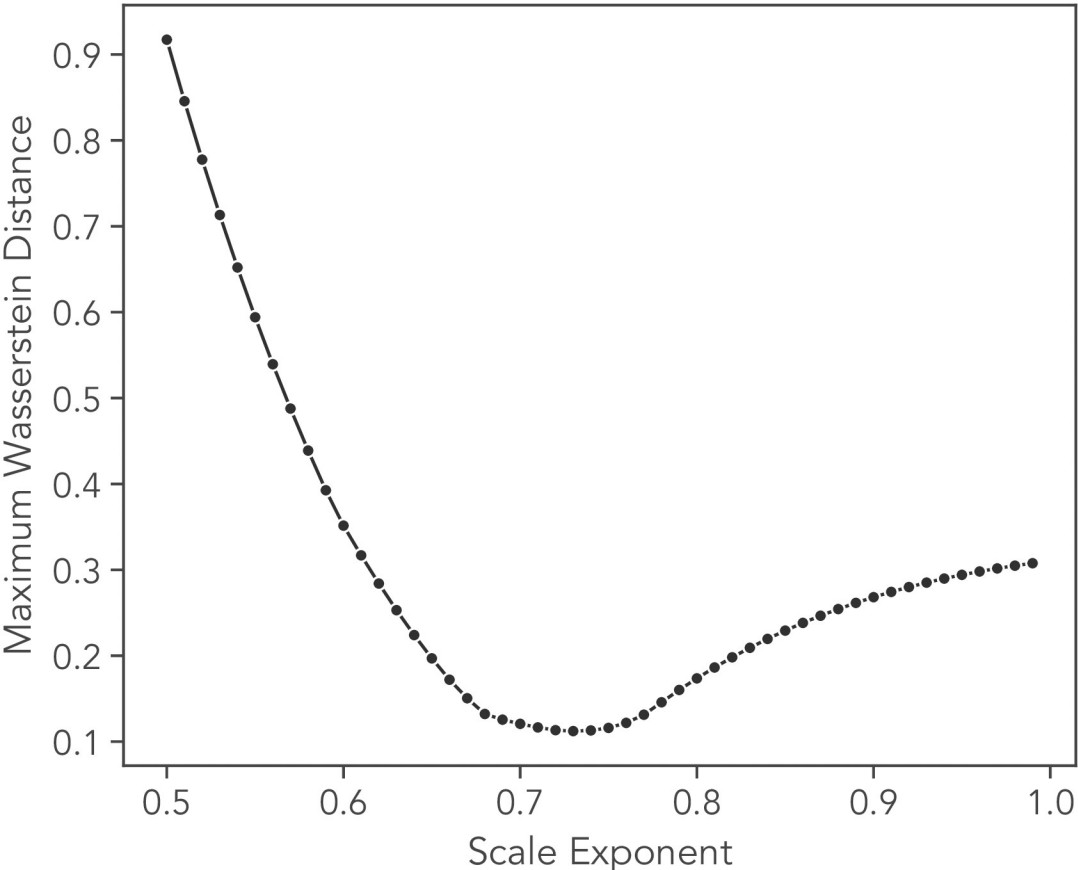

**Fig 6. Optimization of scale exponent using Wasserstein distance.** The maximum value of the Wasserstein distance computed pairwise between the empirical cumulative distribution functions of the re-scaled variable $\eta = \delta x / \Delta^{\beta}$ where $\delta x = x(t + \Delta) - x(t)$, for each lag time $\Delta = \{3, 6, 15, 30, 60, 120\}$ mins used in the analysis. A clear minimum of this statistic is observed at a value of $\beta = 0.73$, which is the value used in Fig 2(b). All cells with track lengths greater than or equal to 120 mins were used in this analysis.

The results of this analysis are shown in Fig 7, with the solid red line representing the MLE fits. The fit for the shape parameter $\gamma$ reveals a spectrum of values all less than one, with a slight increasing trend over time. We also include the maximum likelihood estimates for a Gaussian distribution, plotted as a dashed black curve. Upon inspection, it is clear that the Gaussian distribution provides a poor fit at all values of lag time $\Delta$, while the generalized Gaussian, through incorporation of a shape parameter $\gamma$, seemingly provides the minimal extension necessary for a good fit.

Further to this, we also computed the non-Gaussianity parameter [5]

$$G(\Delta) = \frac{d}{d+2} \times \frac{\overline{\langle \delta^4(\Delta) \rangle}}{\overline{\langle \delta^2(\Delta) \rangle}^2} - 1 \qquad (8)$$

where

$$\overline{\delta^4(\Delta)} = \frac{1}{T - \Delta} \int_0^{T-\Delta} |\mathbf{r}(t + \Delta) - \mathbf{r}(t)|^4 \, dt \qquad (9)$$

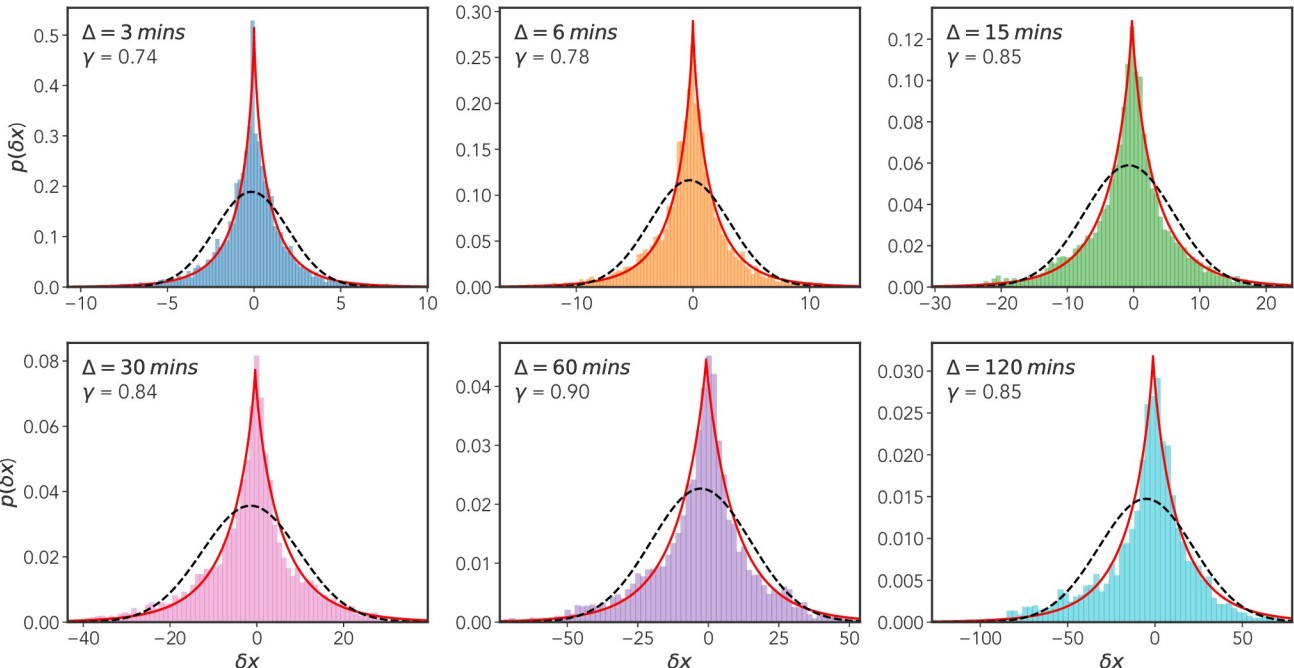

**Fig 7. Non-Gaussian displacement distributions.** $x$-displacement distributions for six different lag times $\Delta = \{3, 6, 15, 30, 60, 120\}$ mins, where $\delta x = x(t + \Delta) - x(t)$ for all time points $t \in [0, T - \Delta]$ where $T$ is the length of the track. Cells included in this analysis have track lengths greater than or equal to 120 mins. Maximum likelihood fits for both a generalized Gaussian distribution (red line) and a standard Gaussian distribution (broken black line) are shown. The shape parameter $\gamma$ for the generalized Gaussian fit is shown along with the lag time $\Delta$ in the top left of each plot.

and $\overline{\delta^2(\Delta)}$ is the TAMSD (4). This quantity is effectively a time-averaged version of the excess kurtosis of a probability distribution, with $G = 0$ for a Gaussian and $G > 0$ for a lepo-kurtotic fat tailed distribution, e.g., a Laplacian distribution. We display the results of this analysis, computed for the data used in Fig 2(a), in Fig 8. We see that for small lag-times there is a pronounced departure from the expected result for a Gaussian distribution, $G(\Delta)$ then decays to approximately zero. Fig 8, together with the results presented in Fig 7, provides strong evidence for the non-Gaussian nature of the underlying dynamical process generating the MPP motion. A notable point is that the convergence of $G(\Delta)$ to the Gaussian value of $G = 0$ occurs prior to the cross over to Fickian diffusion identified in Sec. II B. There has been substantial interest in the converse case, so-called "non-Gaussian yet Fickian diffusion": It is noted in [34] that slowly varying, heterogeneous fluctuations can lead to non-Gaussian displacement distributions, with a time-scale which persists to a time comparable to the cross-over to Fickian diffusion. This is not the case for the MPP dynamics as it appear that the crossover time to Gaussian behavior, $\tau_{gauss}$, (as judged by the decay of $G(\Delta)$ in Fig 8) is around 30 mins, whereas the crossover time to Fickian diffusion $\tau_{fick}$, as discussed later, is around 150 mins.

**3. Fig 2(b) for Y and Z displacements and confinement in the Z-direction.** The curve collapse presented in Fig 2(b), used the re-scaled variable $\eta = \delta x/\Delta^\beta$. Where $\delta x$ represents the $x$-displacement, $\Delta$ the time-lag, and $\beta$ the scale exponent determined via the procedure described above. In Fig 9 we re-plot this for the $y$ and $z$ coordinates, this time using the notation $\eta_y = \delta y/\Delta^{\beta_y}$ and likewise for the $z$-coordinate.

The distributions for the $y$-coordinate are similar to that observed for $\delta x$, with $\beta_y$ estimated to be 0.76. For the $z$ displacement, the result was less convincing, the scale exponent $\beta_z$ was calculated to be 0.59. The reason for this anomaly is likely due to the fact that motion in the $z$-

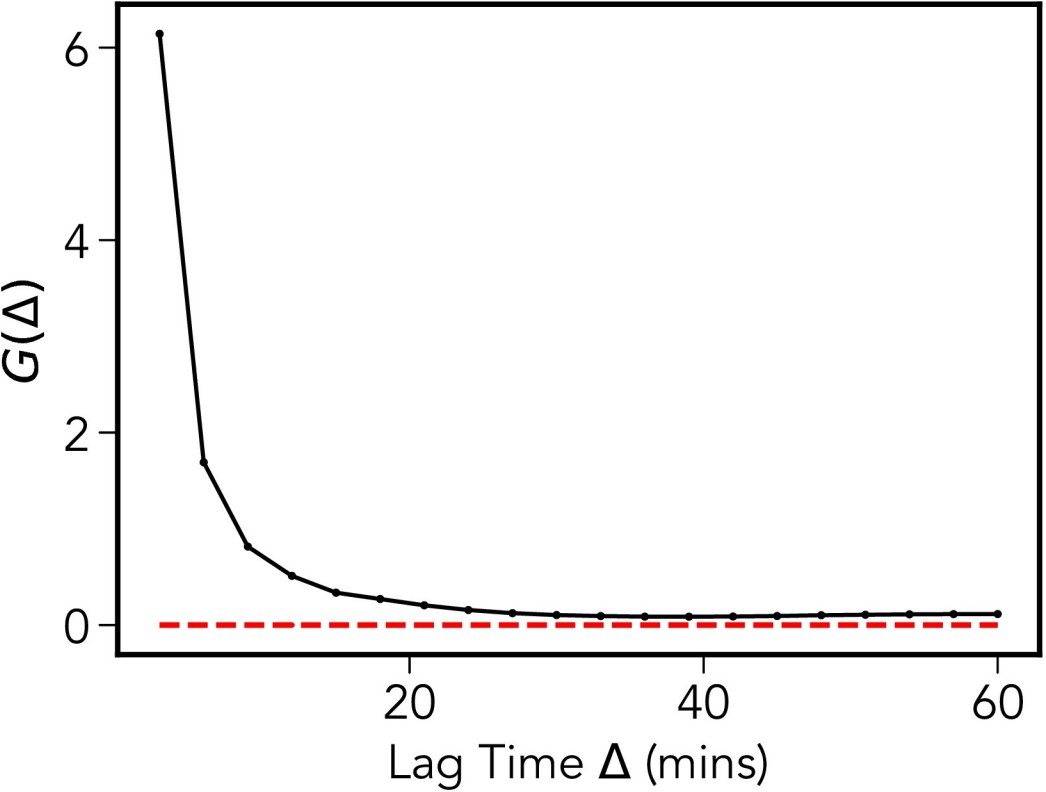

**Fig 8. Non-gaussianity parameter.** The non-Gaussianity parameter $G(\Delta)$ shown for lag-times of up to one hour. At short times the departure from Gaussian behavior is significant, while longer times show a relaxation to the expected result for Gaussian processes $G = 0$.

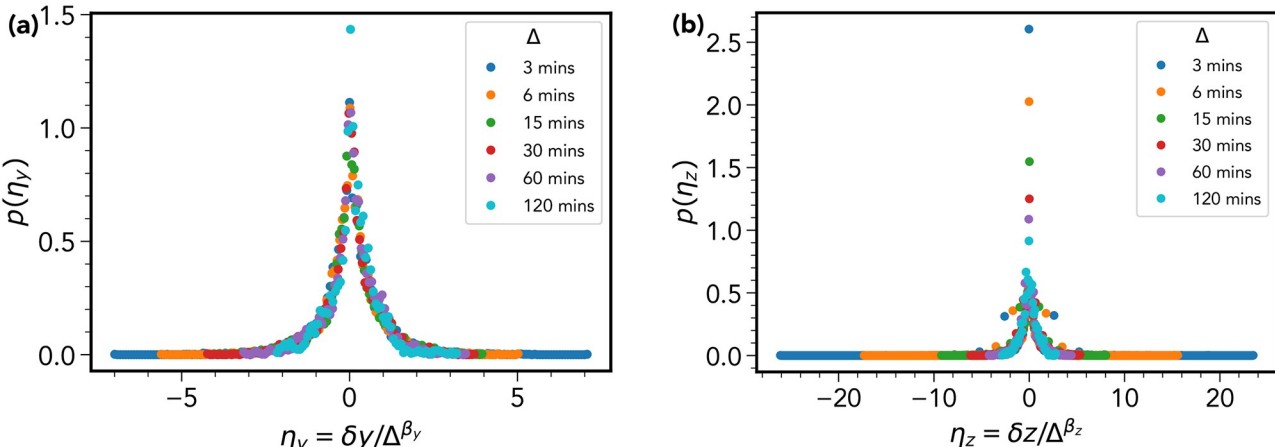

**Fig 9. Curve collapse for y and z displacement distributions.** (a) A repeat of the Fig 2(b) for the re-scaled $y$-displacement $\eta_y = \delta y/\Delta^{\beta_y}$, where we have determined $\beta = 0.76$. Figure (b) shows a the same plot for the z-displacement where this time $\beta_z = 0.59$. The $\eta_y$ distributions reproduce well the result in Fig 2(b), however the same cannot be said of $\eta_z$, which we attribute to sampling bias caused by a limited field of view in the $z$-direction.

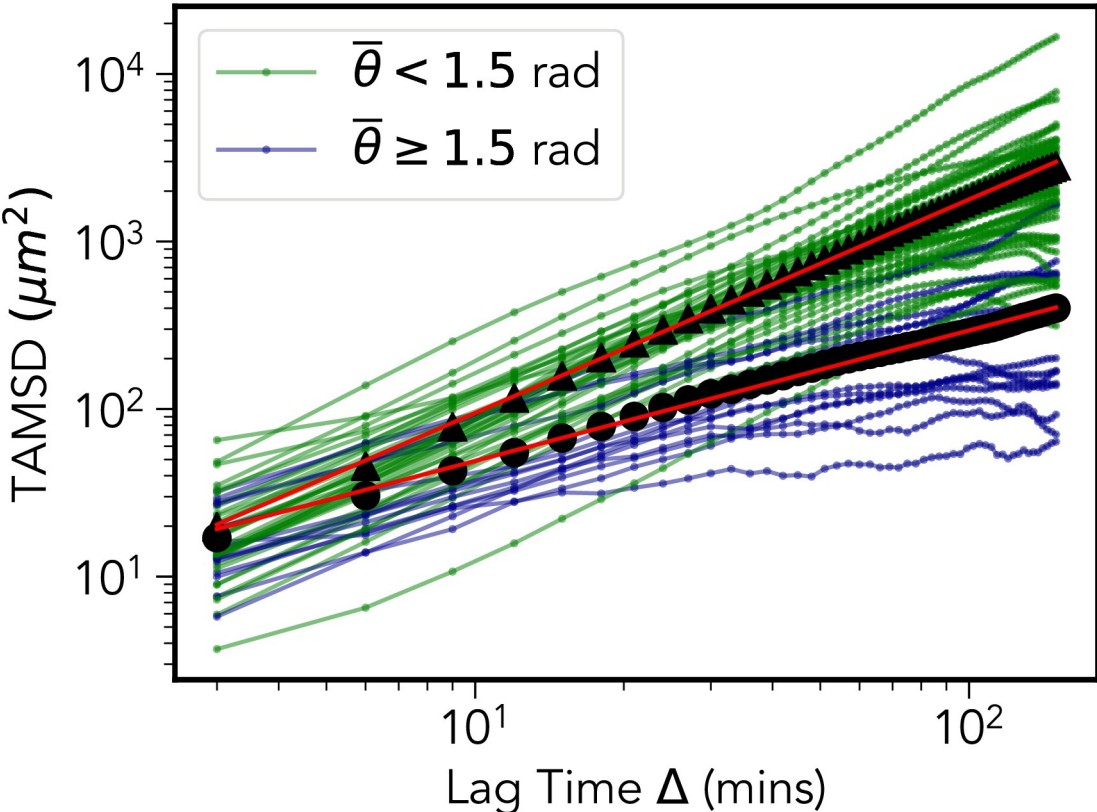

**Fig 10. Time-averaged mean square displacement.** We label the TAMSD of more ($\theta < 1.5$) and less ($\theta \geq 1.5$) persistent cells with green and blue respectively. The corresponding averages over all cells within the more/less persistent groups are shown by the black triangles/circles, along with least-squares fit lines in red which yield exponents of 1.27 for the $\overline{\theta} < 1.5$ group, and 0.78 for $\overline{\theta} \geq 1.5$.

direction appears to be confined. This could be due to the following experimental constraint: There is a limited field of view in the $z$ direction. Light from the confocal microscope is only able to penetrate down to fixed depth within the calvarium, therefore cells possessing a high degree of motility along $z$-axis will leave the field of view early on in the imaging window.

Overall, we note that longer tracks generically belong to one of two categories; 1) motile cells which by chance happen to be moving pre-dominantly in the $xy$-plane, and 2) less motile cells unlikely to be able to leave the field of view during the observation time. We demonstrate this observation in Figs 10 and 11. Dividing the MPP population into two categories based on the trajectory averaged turn angle $\overline{\theta}$ demonstrates that the TAMSD in Fig 2(a) is dominated by a group of 37 (out of 51) persistent cells ($\overline{\theta} < 1.5$) which, as demonstrated in Fig 11(a), remain largely confined to motion within the $xy$-plane during the observation window. This may be contrasted with the other group of 14 less persistent cells, whose mean square displacement is impaired in all directions as shown in Fig 11(b).

### C. Model description

**1. Trajectory coarse-graining.** An arbitrary CTRW may be defined through the specification of three distributions: the run length distribution $f(r)$, the turn angle distribution $g(\theta)$, and the run time distribution $h(\Delta t)$. Here we interpret a run to be undertaken at constant velocity,

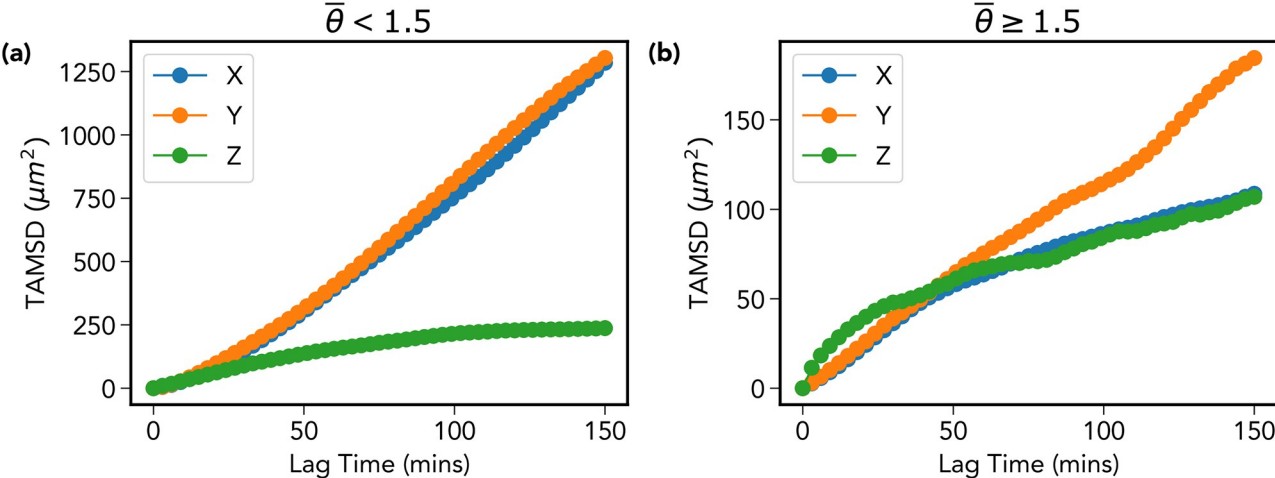

**Fig 11. Confinement in the z-direction.** For all cells used in the TAMSD calculation there exists the appearance of a confinement effect in the *z*-direction due to the limited field of view in this direction. By dividing the cells into two populations based on their average turn angle, we can identify a group of (a) more persistent cells which by chance happen to moving in the *xy*-plane, and (b) generally less motile cells which do not migrate sufficiently in any direction to leave the field of view.

as opposed to a wait-time followed by a discontinuous jump. Many models following this prescription are also known as run-and-tumble (RTB) models, which typically have a Poissonian distributed run-time between subsequent uniformly distributed angular displacements. RTB models are heavily used in the description of the motion of bacteria. The simplest case of such a model would have $r$ and $\Delta t$ related through a fixed speed $v$. For our data-set we will define a variable velocity model using the empirical cumulative distribution functions computed directly from the data.

The empirical distributions are constructed by first applying a coarse-graining transformation to the trajectory. This involves including positions on the trajectory with at least $R$ microns of separation from their previous position, where $R$ is a threshold length-scale determined from the data. Given a track, which is a sequence of position vectors representing the cell centroid $\{\mathbf{r}_t\}$ observed at discrete times $t \in \{0, 1, 2, \ldots, T\}$ we extract a coarse-grained representation, which is an ordered subset of the original $\subseteq \{0, 1, 2, \ldots, T\}$ such that the initial time-point is always included and subsequent points are iteratively found as the first proceeding time point which satisfies the criterion $|\Delta\mathbf{r}_{t\,D(\mathbf{r})} - \Delta\mathbf{r}_{t\,D(\mathbf{r})\,-\,\Delta t}| > R$. Between each pair of subsequent points on the track, we can therefore define a run length $r$ and an associated run time $\Delta t$, and a turn angle $\theta$ as the polar angle between two subsequent run vectors

$$\cos \theta_t = \frac{\mathbf{r}_t \cdot \mathbf{r}_{t+\Delta t}}{|\mathbf{r}_t| \cdot |\mathbf{r}_{t+\Delta t}|}. \tag{10}$$

We show a schematic of this process in Fig 12(a).

This is a heuristic procedure done to remove short-range, transient fluctuations unrelated to genuine movements of the entire cell body. Without this coarse-graining procedure, our simple RTB model will not reproduce the mean square displacement of the experimental trajectories. The Bayesian approach of Metzner et al. [31] provides an elegant way to perform a similar coarse-graining procedure that is adaptive to the data, but our simple cut-off procedure also suffices and has the advantage that it is simple to implement (albeit while losing information pertaining to the short-range movements of the cell body.)

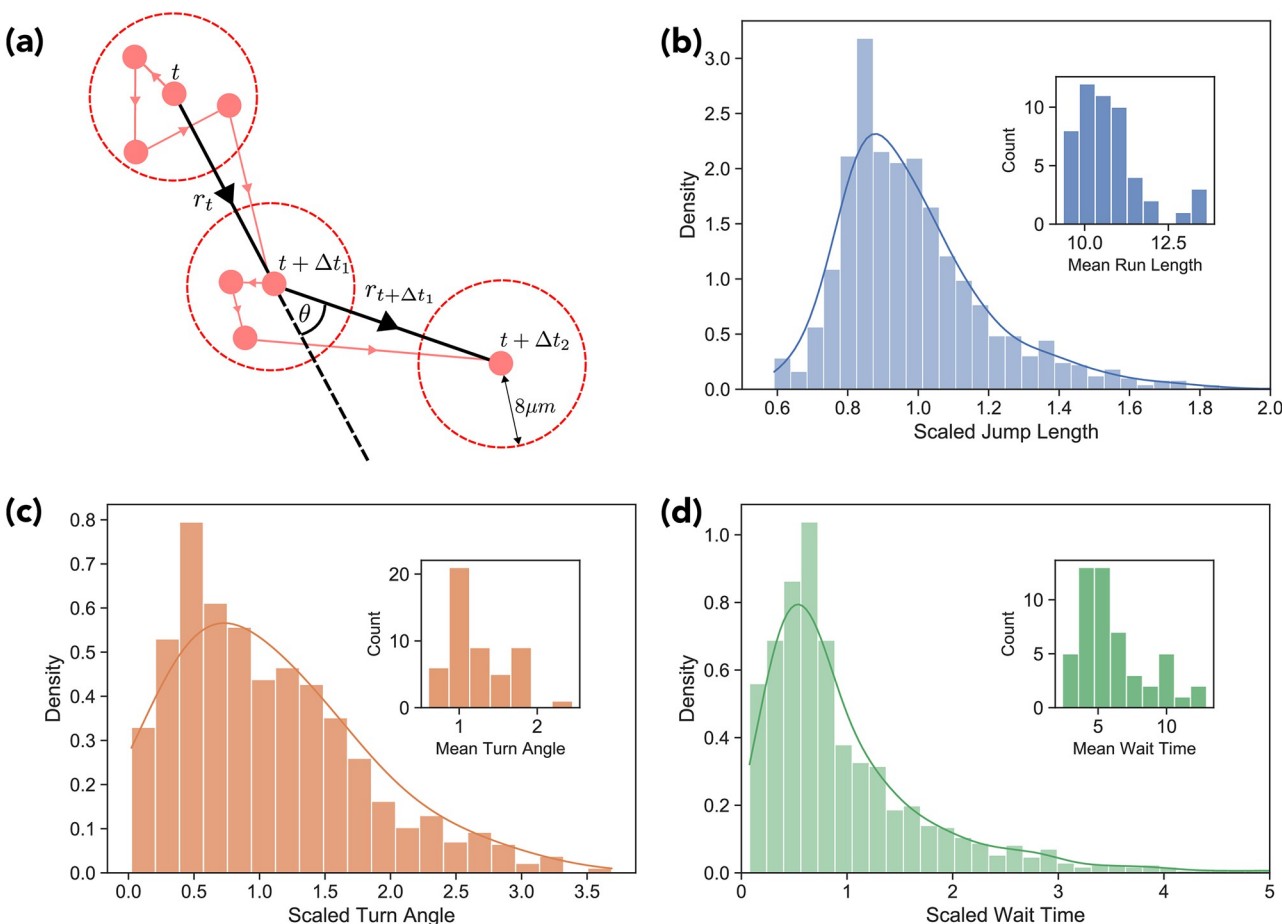

**Fig 12. Coarse-grained trajectory.** a) Schematic of three subsequent points along a coarse grained trajectory. The first point to have at least 7.2 microns displacement from the previous location is included in the coarse-grained trajectory. The distributions of the scaled run lengths (b), scaled turn angles (c), and scaled run time (d) are shown, with their respective distributions of the mean run length per track, etc, shown in the insets.

The threshold value $R$ was determined by minimizing the root sum of the squared error between the simulated and experimental TAMSD.

**2. Empirical distributions.** Using the coarse-grained tracks we are able to construct empirical distributions representing the run-length, run-time, and turn angle distributions required to specify the run and tumble model. However, as previously demonstrated, the MPP trajectories display significant cellular heterogeneity. This observation must therefore be incorporated into our model to faithfully reproduce the experimental statistical analysis. We do this by re-scaling each trajectory's run lengths, turn angles, and wait times by their corresponding mean values $\overline{r}, \overline{\theta}, \overline{\Delta t}$ and aggregating into dimensionless scaled run-length $f(r/\overline{\theta})$, scaled turn angle $g(\theta/\overline{\theta})$ and scaled run-time $h(\Delta t/\overline{\Delta t})$ empirical distribution functions. Histogram plots of the model distributions are shown in Fig 12(b)–12(d).

**3. Model algorithm.** For each simulated cell trajectory we initially assign an empirical mean run-length, run-time and turn angle from one the associated 51 MPP trajectories used in the analysis. The algorithm proceeds by drawing dimensionless samples from the re-scaled empirical distributions, for example $f(r/\overline{r})$, and then multiplying the result with the corresponding mean value $\overline{r}_i$ to obtain the actual value.

As our data analysis and model are three-dimensional, two angles are required to specify a given direction. Assuming a spherical co-ordinate system, we have defined the turn angle $\theta$ between two subsequent runs as the polar angle between the two associated run displacement vectors. We assume no chirality, therefore when updating the position of a simulated trajectory we assign a random azimuthal angle $\phi$.

For each simulated cell trajectory we iteratively generate samples of the three relevant random variables $(r, \theta, \Delta t)$ from the corresponding scaled distributions obtained experimentally (Fig 12(b)–12(d)), and update the position of cell using the scheme described below, until the prescribed temporal track length has been generated. To produce Fig 3 we used linear interpolation to produce a track of simulated positions separated by evenly spaced three minute intervals. This interpolated trajectory could then be used to calculate the TAMSD and compare the model to the empirical data.

**4. Trajectory position update scheme.**   Given a current position $\mathbf{x}_t$ and previous cell position $\mathbf{x}_{t-\Delta t'}$ we wish to obtain the updated position $\mathbf{x}_{t+\Delta t}$ after a run defined by run length $r$, turn angle $\theta$, and run time $\Delta t$. We define run displacement vector $\mathbf{r}_t = \mathbf{x}_t - \mathbf{x}_{t-\Delta t'}$ and associated unit vector $\hat{\mathbf{r}}_t = \mathbf{r}_t/|\mathbf{r}_t|$ which specifies the polar axis of a reference frame centred on the point $\mathbf{x}_t$.

We then use the Gram-Schmidt procedure to obtain two orthogonal unit basis vectors $(\mathbf{e}_1, \mathbf{e}_2)$ that are also orthogonal to $\mathbf{r}_t$. The new position of the trajectory is then

$$\mathbf{x}_{t+\Delta t} = \mathbf{x}_t + r \, \cos \, \theta \hat{\mathbf{r}}_t + r \, \sin(\theta)(cos(\phi)\mathbf{e}_1 + \sin(\phi)\mathbf{e}_2), \tag{11}$$

for an azimuthal angle $\phi \, \theta \, [0, 2\pi)$ drawn from a uniform distribution. This procedure is iterated until the required temporal track length is generated, which in Fig 2(b) is 30000 minutes per track.

**5 Model displacement distributions.**   Fig 3 demonstrates that our model reproduces the scaling of the TAMSD. However, it is also informative to ask to what extent the model reproduces the non-Gaussian displacement distributions observed in the experimental data. In Fig 13 we show a reproduction of Fig 7: the distributions of the variable $\delta x = x(t + \Delta) - x(t)$ for increasing lag times $\Delta$. Interestingly, although the non-Gaussian form is still present—as reflected in the maximum likelihood fits—the distributions become progressively more Gaussian with increasing lag time. This trend does not occur to the same extent in the experimental data: the experimental distribution $p(\delta x; \Delta_{max} = 120 \text{ mins})$ is still distinctly non-Gaussian with a shape parameter $\gamma < 1$. One possible explanation for this discrepancy is that our coarse graining procedure has significantly reduced the likelihood of observing short length displacements at longer lag times. In other words, we have smoothed out the effects of temporal heterogeneity within the track, thus reducing the contribution of the short-length cell centroid fluctuations observed for a given cell during a period of stationarity.

## D. Analysis of heterogeneity using superstatistical Bayesian method

**1. Notation.**   In this section we use the following notation; we denote a collection of random variables $\{X_0, X_1, \ldots, X_T\}$ representing a discrete time stochastic process as $X_{0:T}$ and the corresponding observed values of this process in lower case notation as $x_{0:T}$. For continuous random variables we write the probability density of observing a given realization of the process as

$$p(x_{1:T})\mathrm{d}x_{1:T} = \Pr(X_1 \in (x_1, x_1 + \mathrm{d}x_1) \, , \, \ldots \ldots \, , \, X_T \in (x_T, x_T + \mathrm{d}x_T)) \tag{12}$$

**2. Model specification.**   To demonstrate and quantify the nature of the heterogeneity present within the MPP trajectories we implement the Bayesian procedure of Ref. [31] in which

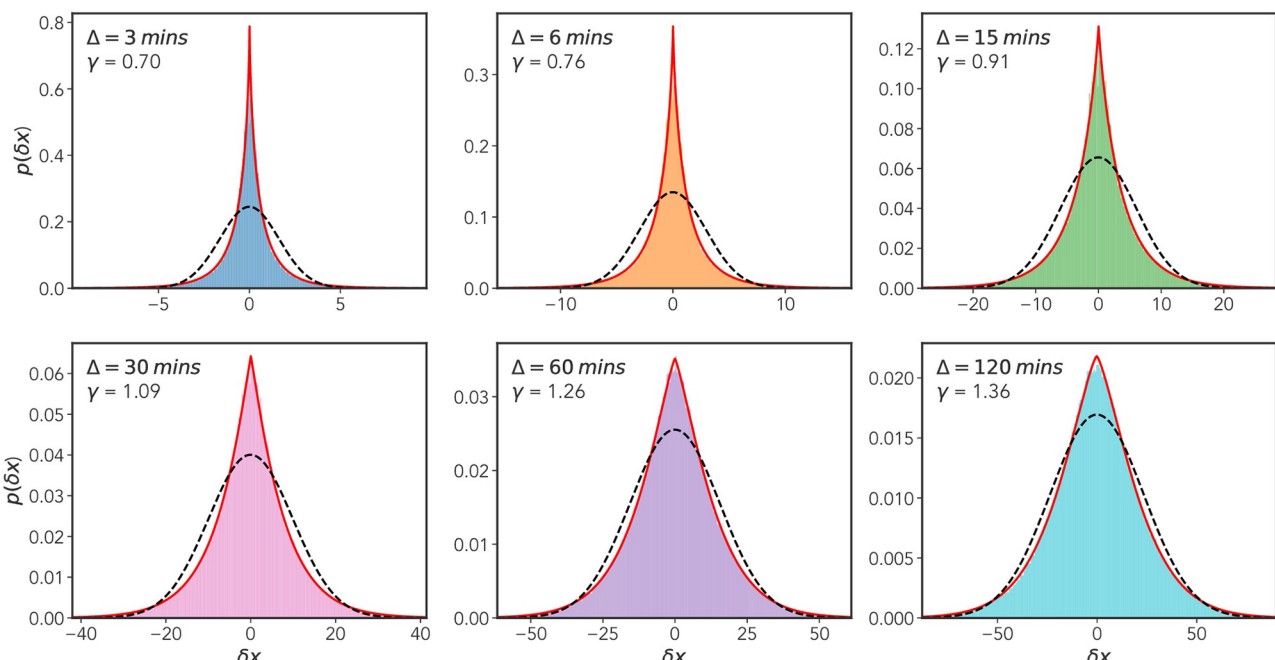

**Fig 13. Model x-displacement distributions.** Reproduction of Fig 7 for the simulated data. As in Fig 7 we show x-displacement distributions for six different lag times Δ = {3, 6, 15, 30, 60, 120} mins, where $\delta x = x(t + \Delta) - x(t)$ for all time points $t \in [0, T - \Delta]$ where $T$ is the length of the track. Each plot shows a maximum likelihood fit for both a generalized Gaussian distribution (red line) and a standard Gaussian (broken black line). The shape parameter $\gamma$ for the generalized Gaussian fit is shown along with the lag time Δ in the top left of each plot. Again in each case we see that the generalized Gaussian provides a superior fit, however, unlike the experimental data the Gaussian provides a progressively better fit with increasing lag time Δ.

cell motion is described hierarchically using a super-statistical model. In this context, the term super-statistics refers to models in which a low-level process representing the dynamics over short spatio-temporal scales, has slowly varying intensive parameters (for example temperature) controlled by a high-level process [48, 49]. It has been demonstrated that the super-position of these two (or more) statistics can lead to systems with stationary states characterized by non-Gaussian fat-tailed probability distributions [50]. The method of Ref. [31], while inspired by this idea, is not completely faithful to it as there is not a strict separation of time-scales between the high level process. To account for rapid changes in motility parameters, regime changes where the cells transitions from a period of inactivity to higher motility, manifested by allowing for abrupt discontinuous jumps in parameter values are allowed. Specifically, Metzner et al. in Ref. [31] describe cell motion using a discrete-time persistent random walk, or AR-1 process

$$\mathbf{v}_t = q_t \mathbf{v}_{t-1} + a_t \mathbf{n}_t \qquad (13)$$

with time varying persistence $q_t \in [-1, 1]$ and activity $a_t \in \mathbf{R}$, and $\mathbf{n}_t$ is a Gaussian noise term with unit variance. These parameters are viewed as random variables evolving according to a time-independent high-level process. This process is modeled through a transform $K$ of the parameter probability distribution.

**3. Bayesian inference algorithm.** The inference algorithm is based on one of the most extensively used mathematical models for analysis of time series with time dependent states—the Hidden Markov Models (HMM) [51]. HMMs consider an observed discrete time-series as

a collection of random variables $Y_{1:T} = \{Y_1, Y_2, \ldots, Y_T\}$ indexed by a time index $t$ whose values are conditionally independent when given the value of a hidden state $X_t$. The correlation structure of the observed time-series is encoded in the hidden state variables as a Markov process.

In our case, the observed variables correspond to the velocity data $\mathbf{v}_{1:T}$, and the hidden state the time-varying parameters $\boldsymbol{\theta}_t = (a_t, q_t)$. We wish to estimate the value of a latent parameter $\boldsymbol{\theta}_t = (a_t, q_t)$ given the entire observed time-series $\mathbf{v}_{1:T}$. Therefore, the inference has a naturally Bayesian interpretation, and is implemented using the forward-backward algorithm for HMMs [51, 52]. In the description of the forward-backward algorithm for simplicity we assume that the probability of a given observation depends only on the current value of the parameter, not on past data points. However, it is readily extended to auto-regressive cases such as that considered by Metzner et al. [31, 53, 54].

As with other Bayesian methods the parameters $\boldsymbol{\theta}_t = (a_t, q_t)$ are viewed as random variables, and the data-set of observed velocities $\mathbf{v}_{1:T}$ are interpreted as fixed. Bayes theorem is used to compute their posterior distribution through multiplication of the prior $p(\boldsymbol{\theta}_t)$—representing previous knowledge of the parameter distribution—with a factorizable likelihood function $p(\mathbf{v}_{1:T}|\boldsymbol{\theta}_t) = \prod_{t=0}^{T} L_t$; where $L_t := p(\mathbf{v}_t|\boldsymbol{\theta}_t)$ is the one-step likelihood function. The normalization factor $p(\mathbf{v}_{1:t})$ is referred to as the model evidence and represents the relative likelihood that the data-set was generated by the model

$$p(\boldsymbol{\theta}_t|\mathbf{v}_{1:T}) = \frac{p(\mathbf{v}_{1:T}|\boldsymbol{\theta}_t) \, p(\boldsymbol{\theta}_t)}{p(\mathbf{v}_{1:T})} \tag{14}$$

for which it is possible to re-write as

$$p(\boldsymbol{\theta}_t|\mathbf{v}_{1:T}) \quad = \quad \frac{p(\mathbf{v}_{1:t}, \mathbf{v}_{t+1:T}|\boldsymbol{\theta}_t) \, p(\boldsymbol{\theta}_t)}{p(\mathbf{v}_{1:T})} \tag{15}$$

$$= \quad \frac{p(\mathbf{v}_{1:t}, \boldsymbol{\theta}_t) \, p(\mathbf{v}_{t+1:T}|\boldsymbol{\theta}_t)}{p(\mathbf{v}_{1:T})} \ . \tag{16}$$

Despite the somewhat cumbersome notation, the forward-backward algorithm results from repeated application of the chain rule for joint probability distributions and the conditional independence relations assumed by the model, in our case Markovian parameter dynamics. Denoting the state space of possible parameter values as $\Theta$

$$p(\boldsymbol{\theta}_t, \mathbf{v}_{1:t}) \quad = \quad \int_{\boldsymbol{\theta} \in \Theta} p(\boldsymbol{\theta}_t, \boldsymbol{\theta}_{t-1}, \mathbf{v}_{1:t}) \mathrm{d}\boldsymbol{\theta}_{t-1} \tag{17}$$

$$= \quad \int_{\boldsymbol{\theta} \in \Theta} p(\mathbf{v}_t, \mathbf{v}_{1:t-1}, \boldsymbol{\theta}_t, \boldsymbol{\theta}_{t-1}) \mathrm{d}\boldsymbol{\theta}_{t-1} \tag{18}$$

$$= \quad \int_{\boldsymbol{\theta} \in \Theta} p(\mathbf{v}_t|\boldsymbol{\theta}_t) p(\mathbf{v}_{1:t-1}, \boldsymbol{\theta}_t, \boldsymbol{\theta}_{t-1}) \mathrm{d}\boldsymbol{\theta}_{t-1} \tag{19}$$

$$= \quad \int_{\boldsymbol{\theta} \in \Theta} p(\mathbf{v}_t|\boldsymbol{\theta}_t) p(\mathbf{v}_{1:t-1}, \boldsymbol{\theta}_{t-1}) p(\boldsymbol{\theta}_t|\boldsymbol{\theta}_{t-1}) \mathrm{d}\boldsymbol{\theta}_{t-1} \tag{20}$$

recognizing that the term $p(\mathbf{v}_{t-1}, \boldsymbol{\theta}_{t-1})$ is the joint parameter and data distribution for the

previous time point $\alpha_{t-1}(\boldsymbol{\theta}_{t-1})$. The function $\alpha$ therefore obeys the following recursion relation

$$\alpha_t(\boldsymbol{\theta}_t) \quad = \quad p(\mathbf{v}_t|\boldsymbol{\theta}_t) \int p(\boldsymbol{\theta}_t|\boldsymbol{\theta}_{t-1})\alpha_{t-1}(\boldsymbol{\theta}_{t-1})\mathrm{d}\boldsymbol{\theta}_{t-1} \tag{21}$$

$$= \quad L_t \; K^F[\alpha_{t-1}(\boldsymbol{\theta}_{t-1})] \tag{22}$$

A similar result may be obtained for the backward recursion for $\beta_t(\boldsymbol{\theta}_t)$ (see [52] for details)

$$\beta_t(\boldsymbol{\theta}_t) \quad = \quad \int_{\boldsymbol{\theta}\in\Theta} p(\boldsymbol{\theta}_{t+1}|\boldsymbol{\theta}_t)p(\mathbf{v}_{t+1}|\boldsymbol{\theta}_{t+1})\beta_{t+1}(\boldsymbol{\theta}_{t+1})\mathrm{d}\boldsymbol{\theta}_{t+1}$$

$$= \quad K^B[L_{t+1}\beta_{t+1}]$$

returning to Eq (15) we are now able to write in the form

$$\text{Posterior} \quad \propto \quad \text{Likelihood} \times \text{Prior}$$

$$= \quad L_t \cdot K^F[\alpha_{t-1}]K^B[L_{t+1}\beta_{t+1}]$$

$$= \quad L_t \cdot Pr_t^F \cdot Pr_t^B$$

where we have represented the integrals over the parameter space as integral transforms $K^F$ and $K^B$ with kernels $p(\boldsymbol{\theta}_t|\boldsymbol{\theta}_{t-1})$ and $p(\boldsymbol{\theta}_{t+1}|\boldsymbol{\theta}_t)$. If the parameter dynamics are time-reversible

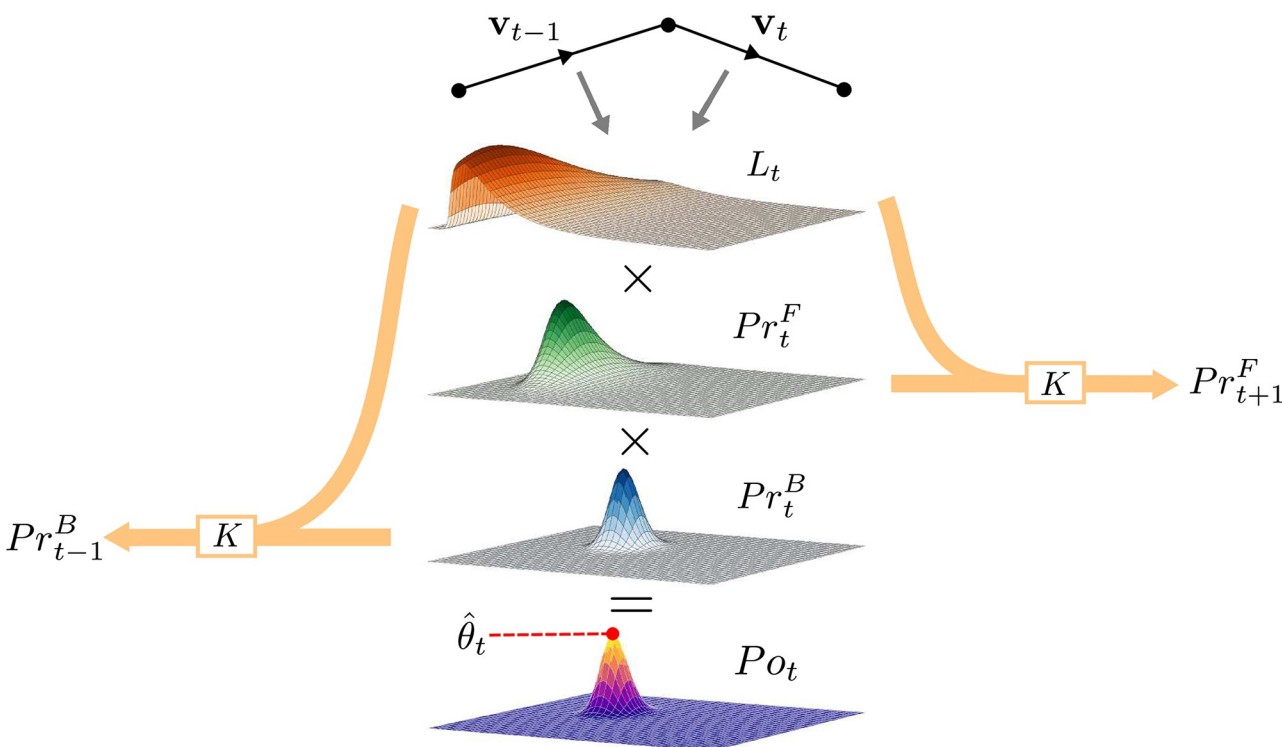

**Fig 14. Schematic of superstatistical Bayesian method.** Given an experimental time-series and an observation likelihood function $L$, which we assume here to be of the form of the Gaussian AR-1 process described in Sec. IVD2, it is possible to compute the posterior distribution of the time-varying activity $a_t$ and persistence $q_t$ parameters using a modified, discretized version of the forward-backward algorithm for hidden Markov models. The posterior parameter distribution is obtained through multiplication of the grids representing the likelihood $L_t$ and the forward and backward priors $Pr_t^F$ and $Pr_t^B$ obtained using the recursion relations presented in Sec. IVD3. Point parameter estimates $\hat{\theta}_t$ can be obtained from this posterior, typically through the mean or the mode. The subsequent priors in the forward and backward direction are computed independently using the ad-hoc transformation $K$.

then $K^F = K^B$. We can therefore view the prior as the product of two independently obtained forward and backward priors $Pr^F$ and $Pr^B$ which have incorporated all information from the data in both directions of time converging onto time $t$.

**4. Adapted implementation of Metzner et al.** Metzner et al. used a grid-based implementation of the above algorithm [31], in which the parameter state space $\Theta$ is discretized over a $200 \times 200$ rectangular grid. Starting from a uniform prior the forward and backward recursions are run independently from time-points $t + 2$ and $T$ respectively. The parameter bounds are $q_t \in (-1, 1)$ and $a_t \in (0, a_{max})$ where the limit $a_{max}$ is data-set dependent. Apart from the discrete approximation which facilitates computational efficiency and allows for a direct estimation of the model evidence; the key advantage of this technique is to generalize the integral transform $K$ of the forward and backward priors $\alpha$ and $\beta$ to allow for a more general class of transformations. In cases where there is no prior knowledge regarding the form of the parameter dynamics, it is preferred to keep the transformation $K$ as general as possible, as to encapsulate both the gradual and potentially abrupt parameter changes, while not relying on a specific functional form. This process is shown graphically in Fig 14. The likelihood function $L_t$ is approximated directly from the data and multiplied with priors. Following Ref. [31] we use a two step process to transform the parameters; firstly, to account for possible abrupt parameter

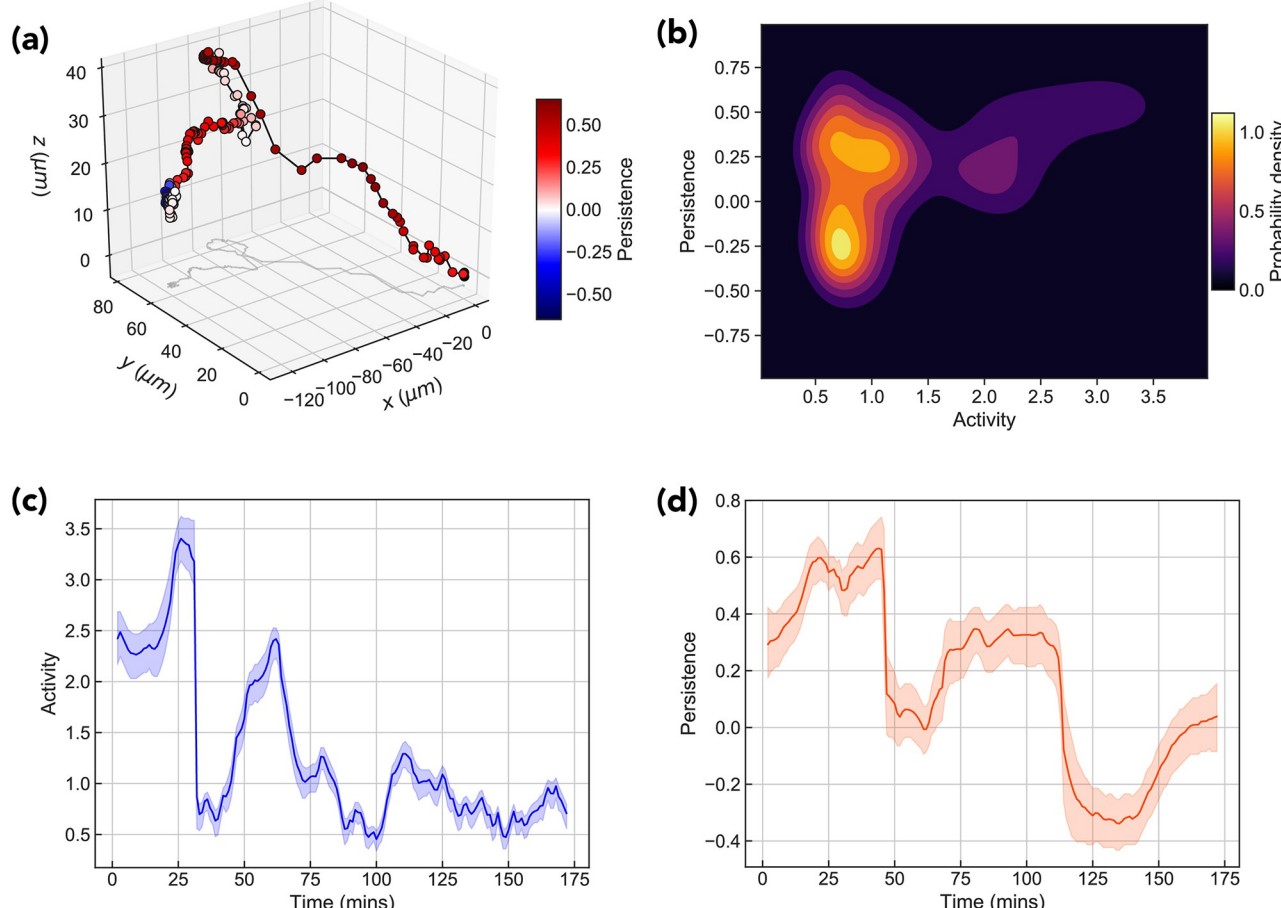

**Fig 15. Results of Bayesian analysis for example trajectory.** (a) Shows an example MPP trajectory color coded according to the value of the persistence parameter $q$ at that time. Figure (b) shows the time-averaged mean posterior distribution. Figures (c) and (d) show the time evolution of the posterior mean of the activity parameter $a$ and persistence parameter $q$ respectively.

changes, a minimum probability $p_{min}$ is assigned to each point of the grid. Secondly, to account for gradual changes in parameter values, we apply a uniform convolution to the parameter grid with box of radius $R$.

**5. Results of Bayesian analysis.** To fix the value of the hyper-parameters $p_{min}$ and $R$ controlling the high-level parameter transformation, we select the values of $p_{min}$ and $R$ which maximize the model evidence $p(\mathbf{v}_{1:T})$, evaluated by summing the (un-normalized) posterior density over all points of the grid. The model evidence provides a quantitative measure of the likelihood that the data was generated by the model. The discrete sum approximates the integral

$$p(\mathbf{v}_{1:T}|p_{min}, R) = \int_{\boldsymbol{\theta} \in \Theta} p(\boldsymbol{\theta}_{1:T}, \mathbf{v}_{1:T}|p_{min}, R) \approx \sum_{ij} p(\boldsymbol{\theta}_{ij}, \mathbf{v}_{1:T}|p_{min}, R) \cdot \Delta_{\boldsymbol{\theta}} \qquad (23)$$

where we use the indices $ij$ to represent the points of the discretized parameter grid and $\Delta_{\boldsymbol{\theta}}$ represents the voxel size of the grid. We also note that the results presented herein are not overly sensitive to variation of these hyper-parameters, however, we use the model evidence maximum maximization criteria as means to fix their value. This is a form of what is known as empirical Bayes in the mathematical statistics literature.

In Fig 15 we show the result of this analysis applied to a particular trajectory from the ensemble of MPP tracks. For each posterior $Po_t$ we compute the means of both the activity and persistence parameters, along with their 50 percent credible regions, shown in Fig 15(c) and 15(d). In Fig 15(a) we annotate a trajectory according to the inferred persistence parameter

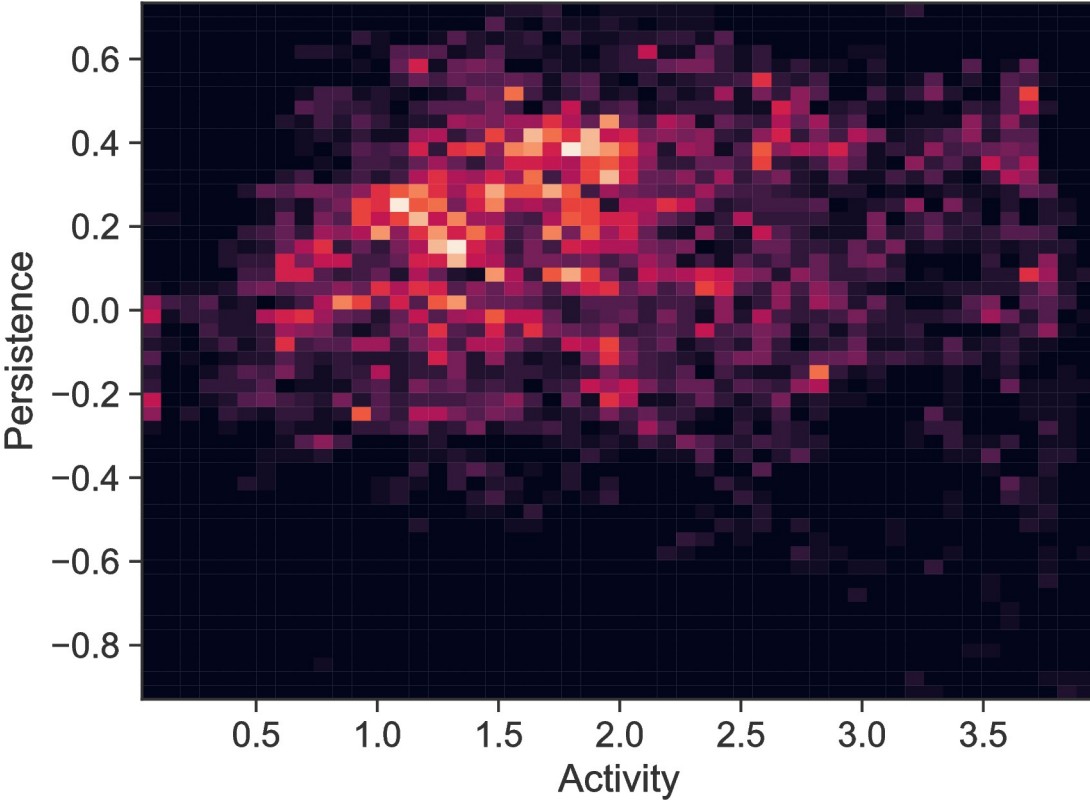

**Fig 16. Results of Bayesian analysis for entire ensemble of MPPs.** 2D histogram plot of time varying activity and persistence parameters $a_t$ and $q_t$ collated over all MPP tracks.

values. Visually, it is clearly seen that the algorithm is able to pick out periods of persistent motion interspersed with periods of low activity and persistence. The time-averaged posterior distribution shown in Fig 15(b) provides further insight into the motion of cell over the entire trajectory; for this particular track we see that there exists three distinct modes, one low activity anti-persistent, one low-activity persistent, and one high activity.

Repeating this analysis for the entire ensemble of MPP tracks, we observe a large spread in the values of the $a_t$ and $q_t$, shown in the histogram plot in Fig 16.

## Author Contributions

**Conceptualization:** Benjamin Partridge, Cristina Lo Celso, Chiu Fan Lee.

**Data curation:** Sara Gonzalez Anton, Reema Khorshed, George Adams, Constandina Pospori.

**Formal analysis:** Benjamin Partridge.

**Funding acquisition:** Cristina Lo Celso, Chiu Fan Lee.

**Investigation:** Benjamin Partridge, Sara Gonzalez Anton, Reema Khorshed, George Adams, Constandina Pospori, Chiu Fan Lee.

**Methodology:** Benjamin Partridge, Chiu Fan Lee.

**Project administration:** Cristina Lo Celso, Chiu Fan Lee.

**Resources:** Cristina Lo Celso.

**Software:** Benjamin Partridge.

**Supervision:** Cristina Lo Celso, Chiu Fan Lee.

**Writing – original draft:** Benjamin Partridge.

**Writing – review & editing:** Benjamin Partridge, Cristina Lo Celso, Chiu Fan Lee.

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
