## [Decision Letter · Decision Letter 0]

4 May 2022

PONE-D-22-03550

Heterogeneous run-and-tumble motion accounts for transient non-Gaussian super-diffusion in haematopoietic multi-potent progenitor cells

PLOS ONE

Dear Dr. Lee,

Thank you for submitting your manuscript to PLOS ONE. After careful consideration, we have decided that your manuscript does not meet our criteria for publication and must therefore be rejected.

Specifically:

PLOS ONE requires that reported research meets all applicable standards for the ethics of experimentation and research integrity (http://journals.plos.org/plosone/s/criteria-for-publication#loc-6).

Upon consultation with the journal Editorial Board, we have concluded that the method of euthanasia used in this study (“Recipient mice are lethally irradiated using two doses of 5.5Gy of γ-radiation three hours apart”) was not justified in this case.

We also note that the UK Home Office ‘Guidance on the operation of the Animals (Scientific Procedures) Act 1986’ found at https://www.gov.uk/guidance/guidance-on-the-operation-of-the-animals-scientific-procedures-act-1986 defines ‘irradiation […] with a lethal dose without reconstitution of the immune system, or reconstitution with production of graft versus host disease’ as ‘Severe’ with regards to the degree of pain, suffering, distress or lasting harm expected to be experienced by an individual animal during the course of the procedure. Please refer to Appendix G (pp 118-121, specifically Section III 3(d)).

As such, while we have received a positive report from an independent reviewer regarding other aspects of this study, we cannot in light of these concerns consider your study further for publication and are rejecting your manuscript.

In future work, we strongly encourage you to carefully consider the use of an acceptable method for euthanasia. Please note that we reserve the right to reject any submission that does not meet our standards for the ethics of experimentation, which in some cases may be more stringent than local ethical standards.

I am sorry that we cannot be more positive on this occasion, but hope that you appreciate the reasons for this decision.

Kind regards,

Haroldo V. Ribeiro

Academic Editor

PLOS ONE

Reviewers' comments:

Reviewer's Responses to Questions

**Comments to the Author**

1. Is the manuscript technically sound, and do the data support the conclusions?

Reviewer #1: Yes

2. Has the statistical analysis been performed appropriately and rigorously? 

Reviewer #1: Yes

3. Have the authors made all data underlying the findings in their manuscript fully available?

Reviewer #1: Yes

4. Is the manuscript presented in an intelligible fashion and written in standard English?

Reviewer #1: Yes

5. Review Comments to the Author

Reviewer #1: This work is focused on the statistical analysis of the cell trajectory obtained from 3D in vivo images of haematopoetic multi-potent pogenitor (MPP) cells in the irradiated bone marrow cavity of murine calvaria. The authors clearly and exhaustively demonstrate that cells exhibit transient non-Gaussian super-diffusion over time-scales of biological interest. To support this finding, the authors use a normally successful run-and-tumble model (RTM) for analyzing bacterial movement to account for heterogeneity in ensemble dynamics. The fundamental thesis of the work is that the incorporation of heterogeneity into the RTM model is a necessary and sufficient condition to explain the non-Gaussian character of super-diffusive behavior.

In the approach proposed by the authors, by means of temporal extrapolation, it is possible to find the crossover time for which the diffusion process underlying cell movement transitions towards a normal diffusion process, ceasing to be anomalous.

The authors make an essential statement from the point of view of the physics and biology of these systems: they say that, from these estimates of the parameters involved in the different stages of the movement of these cells, essential information could be obtained to help in the understanding of how the MPP mobility could influence blood cell generation regulations.

The work is very well written. The methodology is discussed in an understandable and quite complete manner. The materials and methods are presented in a clear and objective way, except for the very summary description of the set-up used for microscopy, assuming in the reader a previous understanding of the importance of laser confocal microscopy (LSCM) that may not be so obvious. In fact, the importance of fluorescence in the methodology cannot be neglected, as it is a crucial part of data collection. Statistical methods are discussed and applied with great competence and extensive knowledge on the part of the authors. There is a wealth of observations and technical subtleties in the work that can be considered admirable. The work is inserted with value and grandeur in a vast literature and very well mastered by the authors, as can be seen from the abundant and useful bibliography.

Let me consider, briefly and respectfully, the only weak point that I see in the whole work. While acknowledging that this is essentially a statistical work, I would have liked to have seen a broader discussion of the scientific significance of an anomalous diffusion process in this context. Let me explain myself better: the data processing was done in a very competent and comprehensive way precisely to establish the anomalous nature of the diffusive process observed in cellular movement - which was clearly demonstrated, through a broad, rigorous statistical analysis, and conducted with competence and richness of detail. The meaning of all this, however, although outlined in the discussions conducted by the authors, perhaps deserves further exploration. After all, we use all these methods to analyze and interpret the meaning of the data. My impression is that in the "analysis" question, the work was left, but in the "interpretation" question, the work still owed a little more effort.

Anyway, it is, on the whole, a work of great value and of great importance in its field.

Apart from the occasional comma separating subject from verb in different places of the text, which can be easily corrected in a last and careful reading, it is possible that, on page 4, in the step 2 of the procedure, instead of "actual run-time" the authors wanted say "current run-length".

In summary, the work is scientifically sound, technically correct, and represents a very welcome contribution to a vast area of research, potentially being very stimulating and useful to a large audience and a solid scientific community.

6. PLOS authors have the option to publish the peer review history of their article (what does this mean?). If published, this will include your full peer review and any attached files.

Reviewer #1: No

- - - - -

---

## [Editor Report · Decision Letter 1]

22 Jul 2022

Heterogeneous run-and-tumble motion accounts for transient non-Gaussian super-diffusion in haematopoietic multi-potent progenitor cells

PONE-D-22-03550R1

Dear Dr. Lee,

We’re pleased to inform you that your manuscript has been judged scientifically suitable for publication and will be formally accepted for publication once it meets all outstanding technical requirements.

Kind regards,

Haroldo V. Ribeiro

Academic Editor

PLOS ONE

Additional Editor Comments (optional):

I thank the authors for adequately addressing the minor points of our reviewer and for all the clarifications related to animal work raised by our internal editors. When preparing the final document, please remove the word "lethally" that appears strikethrough in section "Mice" of Materials and Methods.
---

## [Editor Report · Acceptance letter]

2 Sep 2022

PONE-D-22-03550R1 

Heterogeneous run-and-tumble motion accounts for transient non-Gaussian super-diffusion in haematopoietic multi-potent progenitor cells 

Dear Dr. Lee:

I'm pleased to inform you that your manuscript has been deemed suitable for publication in PLOS ONE. Congratulations! Your manuscript is now with our production department. 

Kind regards, 

on behalf of

Dr. Haroldo V. Ribeiro 

Academic Editor

PLOS ONE